# The association between dementia severity and hospitalisation profile in a newly assessed clinical cohort: the South London and Maudsley case register

Usha Gungabissoon  ,[1,2] Gayan Perera  ,[2] Nicholas W Galwey,[3] Robert Stewart[2,4]

¹Epidemiology (Value Evidence and Outcomes), GSK, Brentford, UK
²Department of Psychological Medicine, Institute of Psychiatry, Psychology & Neuroscience, King's College London, London, UK
³Target Sciences, GSK, Stevenage, UK
⁴NIHR Maudsley Biomedical Research Centre, South London and Maudsley NHS Foundation Trust, London, UK

**Correspondence to**
Usha Gungabissoon;
usha.2.gungabissoon@gsk.com

## ABSTRACT

**Objectives** To evaluate the risk and common causes of hospitalisation in patients with newly diagnosed dementia and variation by severity of cognitive impairment.

**Setting** We used data from a large London mental healthcare case register linked to a national hospitalisation database.

**Participants** Individuals aged ≥65 years with newly diagnosed dementia with recorded cognitive function and the catchment population within the same geography.

**Outcome measures** We evaluated the risk and duration of hospitalisation in the year following a dementia diagnosis. In addition we identified the most common causes of hospitalisation and calculated age-standardised and gender-standardised admission ratios by dementia severity (mild/moderate/severe) relative to the catchment population.

**Results** Of the 5218 patients with dementia, 2596 (49.8%) were hospitalised in the year following diagnosis. The proportion of individuals with mild, moderate and severe dementia who had a hospital admission was 47.9%, 50.8% and 51.7%, respectively (p= 0.097). Duration of hospital stay increased with dementia severity (median 2 days in mild to 4 days in severe dementia, p 0.0001). After excluding readmissions for the same cause, the most common primary hospitalisation discharge diagnoses among patients with dementia were urinary system disorders, pneumonia and fracture of femur, accounting for 15%, 10% and 6% of admissions, respectively. Overall, patients with dementia were hospitalised 30% more than the catchment population, and this trend was observed for most of the discharge diagnoses evaluated. Standardised admission ratios for urinary and respiratory disorders were higher in those with more severe dementia at diagnosis.

**Conclusions** Individuals with a dementia diagnosis were more likely to be hospitalised than individuals in the catchment population. The length of hospital stay increased with dementia severity. Most of the common causes of hospitalisation were more common than expected relative to the catchment population, but standardised admission ratios only varied by dementia stage for certain groups of conditions.

## Strengths and limitations of this study

► We evaluated hospitalisation in a large and representative sample of newly diagnosed cases of dementia in a London mental healthcare case register linked to a national hospitalisation database providing near-complete outcome ascertainment.

► Severity of dementia at the time of diagnosis was determined from the Mini-Mental State Examination (MMSE) score or Health of the Nation Outcome Scales impairment score, which has acceptable/good psychometric properties and correlates with MMSE score measurement.

► We obtained the analysed samples from a single service provider in a single London catchment, although highly socially diverse, which may limit the generalisability of our findings to other settings.

► The analyses did not account for comorbidities, use of medications, institutional residence or socioeconomic status, aetiology of dementia or hospitalisation rates prior to the diagnosis of dementia, all of which may influence the risk of hospitalisation.

## INTRODUCTION

In the United Kingdom, there are around 850 000 people living with dementia and the associated annual healthcare costs are around £4.3 billion; major contributing factors to costs include disease severity and hospitalisation.[1] It follows that the associated healthcare costs are expected to increase in line with the increasing prevalence of dementia as a result of population ageing.[2] Compared with age-matched controls, people with dementia are more likely to require an acute hospital admission,[3 4] more hours of care and longer hospital stays.[5] Hospitalisation also often represents a pivotal event for people with dementia due to increased risk of admission to long-term institutional care, functional decline, mortality, loss of independence and impact

on caregivers.[6] To date most published studies examining hospital admissions of people with dementia have evaluated prevalence samples of people with dementia.[7–9] Other studies have been small,[3 10 11] or have not taken into account the severity of dementia.[3 8 9 12] Some have ascertained admission information from family carers[10 13] which is prone to recall bias.

There is increasing focus on improving interventions and causes of hospitalisation which may be prevented by optimised primary care in the UK for those with dementia.[14] Identifying common causes of hospitalisation in this population could help guide development of preventive strategies to reduce the risk of unnecessary events and/or to avoid lengthy and disruptive admission episodes. However, it is highly likely that the profile of hospitalisations and underlying causal pathways will vary according to the severity of dementia, given the substantial changes that occur during the course of the disease in level of frailty and ability to self-care, not to mention direct effects of neuropsychiatric symptoms on physical health risk. Despite the impact of hospitalisation and the potential for preventative intervention, we are not aware of any previous study that has evaluated the risk and identified the most common causes of hospital admission in newly diagnosed dementia and the extent to which this varies with dementia severity.

In this study, we sought to evaluate the frequency and common causes of hospitalisation in a newly diagnosed cohort of people with a dementia diagnosis relative to the catchment population, and also the potential association between this relative likelihood of cause-specific hospitalisation and severity of cognitive impairment, using a large cohort drawn from a mental healthcare case register linked to national hospitalisation episodes.

## METHODS
### Study design and setting
This retrospective observational cohort study involved analysis of newly presenting patients who received a dementia diagnosis. We evaluated hospital episodes in the 12-month period following this index diagnosis date.

The Clinical Record Interactive Search (CRIS) data resource was used to identify dementia cases. CRIS provides research access to anonymised electronic health records (EHR) from the South London and Maudsley NHS Foundation Trust (SLaM). SLaM provides mental healthcare, including dementia assessment and management, for a south London catchment containing approximately 1.2 million residents; EHRs were implemented across all SLaM services from 2006.[15] CRIS data are linked to mortality records and national Hospital Episode Statistics (HES) described below. The Oxfordshire Research Ethics Committee C (reference 18/SC/0372) has approved CRIS at the Maudsley as a data resource for secondary analysis.

Routine diagnoses recorded in SLaM are structured according to the WHO International Classification of Diseases, 10th edition (ICD-10) and are supplemented in CRIS, by a natural language processing (NLP) algorithm ascertaining diagnoses recorded in correspondence and other text fields.[15 16] Dementia was defined on the presence of F00*, F01*, F02*or F03* ICD diagnosis codes recorded up to the sixth position in the structured data, or F00,F01, F02, F03 dementia, Alzheimer's disease, vascular dementia and mixed dementia from NLP. Hospital outcomes (hospital admissions and its cause) for patients with dementia were obtained from the HES–CRIS linkage. HES contains details of all inpatient admissions at NHS hospitals in England.[17] Discharge diagnoses are recorded as ICD-10 codes and are available for each hospitalisation episode. Additionally, a subset of the HES database detailing hospitalisations for all residents within SLaM's catchment was used in this analysis to generate expected rates for standardisation.

### Study participants
We aimed to identify people with newly diagnosed dementia and retrieved records from CRIS for patients with a first recorded diagnosis of dementia between 1 January 2008 and 31 December 2012. We restricted the sample to individuals aged ≥65 years at the time of dementia diagnosis, to those with a measure of cognition within 6 months of their dementia diagnosis, and to those with a Mini-Mental State Examination (MMSE) score of <28 or Health of the Nation Outcome Scales (HoNoS) cognitive impairment score >1. Patients who were receiving care from an acute hospital liaison service at the time of the initial diagnosis were excluded since they reflected cases whose dementia diagnosis might have been precipitated by a hospitalisation and who would have an accompanying higher risk of further events.

### Covariates
Age at diagnosis (in 5 year bands), sex, ethnicity and dementia severity (within 6 months of dementia diagnosis) were extracted from CRIS. Dementia severity was estimated primarily from the MMSE score recorded closest to the dementia diagnosis date. If no MMSE score was present within 6 months of the diagnosis date, the closest-recorded cognitive impairment subscale of HoNOS was used if data on this were available within the 6-month period around dementia diagnosis. HoNOS is a clinician-rated instrument usually completed at first assessment, which contains subscales rated 0 (no problem) to 4 (severe or very severe problem), has acceptable/good psychometric properties and correlates with MMSE measurement.[18 19] Dementia severity was categorised as mild, moderate or severe based on MMSE scores of 21–27, 10–20 and 0–9 which is similar to cut-offs used by the National Institute for Health and Care Excellence (NICE), or a HoNOS cognitive impairment subscale score of 2, 3 or 4, respectively.[20]

All acute general hospital inpatient admissions (both elective and unplanned) for cases were obtained for the 12-month period following the date of dementia diagnosis.

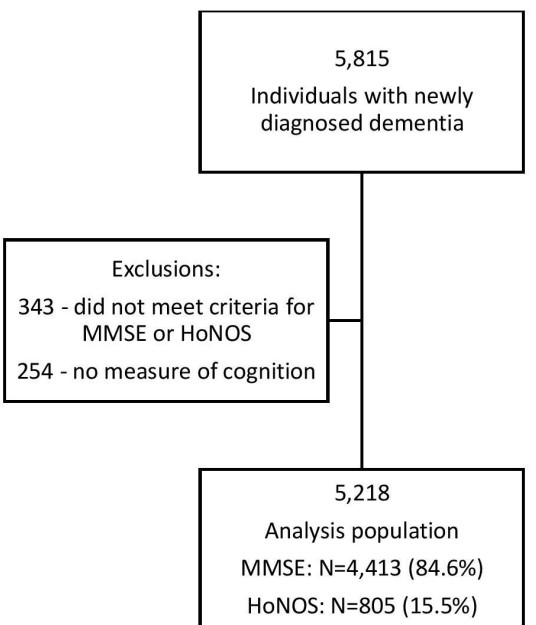

**Figure 1** Identification of the case sample. HoNoS, Health of the Nation Outcome Scales; MMSE, Mini Mental State Examination.

Follow-up was censored at the earliest of 31 March 2013 or date of death. Hospitalisations were defined from HES episodes, combining contiguous episodes (ie, where start and end dates were on the same day). Length of hospital stay (LOS) was defined as the number of days from admission to discharge. The three-digit primary ICD-10 discharge diagnosis was obtained for cases and the catchment population for each hospitalisation. All diagnoses were also grouped at the highest level (letter) ICD-10 code, in line with ICD-10 chapter classifications, broadly related to the body system affected.

Hospitalisations for the catchment population from January 2008 to March 2012 were extracted, and the age and sex profile of the catchment was obtained from the UK Census data for the same time frame. Figure 1 shows the derivation of the case sample.

## Statistical analyses

Microsoft Excel for Office 365 ProPlus and Stata V.13.0 were used for analyses. Following a descriptive analysis, we calculated the cumulative incidence of general hospital admissions in the dementia cohort accounting for person-years (py) of follow-up. The association of dementia severity with risk of hospitalisation and length of stay was evaluated by $\chi^2$ test and Kruskal-Wallis tests, respectively.

Age and sex standardised admission ratios (SARs) with 95% CI were calculated overall and for each of the 20 most frequent causes of hospital admission for the 12-month period from the dementia diagnosis based on the three-digit ICD-10 codes. The SAR was defined as the observed number of cases in the dementia cohort admitted with a given cause, divided by the number expected, the second estimate derived from age-specific and sex-specific rates for admissions with that cause in the SLaM catchment. SARs were also calculated for diagnoses at ICD-10 chapter level. SARs were then calculated separately for each category of dementia severity at the time of dementia diagnosis.

To describe variation in the SAR for each specific cause of admission of interest (three-digit ICD-10 code or ICD-10 chapter) by severity of dementia at diagnosis, linear regression models, with SAR as the dependent variable and dementia stage (mild, moderate or severe) as the exposure variable, were fitted. The slope of the regression (direction and magnitude) was used to describe the trend across dementia stages for each specific cause as a way in which to quantify whether SARs tended to increase or decrease with dementia severity. In a sensitivity analysis, we excluded repeat admissions (defined as repeat admissions for the same three-digit ICD-10 code in the 12-month period of interest).

## RESULTS

The mean age (SD) of the included patients in the case sample (n=5218) was 82.2 (7.0) years, and almost two-thirds were women (n=3338; 64.0%); descriptive characteristics of the analysed cohort by the severity of dementia are displayed in table 1. At the time of dementia diagnosis, 39.4% were classified as mild, 50.1% as moderate and 10.4% as severe. The 12-month mortality rate from dementia diagnosis was 15.6% overall: 10.9% for mild severity, 17.5% for moderate and 24.6% for severe. Approximately half of the patients with dementia were hospitalised during this period (table 1). While the mean number of admissions did not differ substantially between the dementia severity groups, the median duration of inpatient stay (for all admissions in the 12-month period) increased with higher severity of dementia at diagnosis (p 0.0001). Although the proportion of patients with at least one admission increased with dementia severity, this did not reach statistical significance (p 0.097). Patients with dementia were hospitalised 30% more than the catchment population (SAR: 1.3, 95% CI 1.2 to 1.3) during the 12-month follow-up and this SAR was similar across dementia severities: mild 1.2 (95% CI 1.2 to 1.2), moderate 1.3 (95% CI 1.3 to 1.3) and severe 1.3 (95% CI 1.2 to 1.3).

The 20 most common three-digit primary discharge diagnoses for hospitalisation episodes in the year following dementia diagnosis are shown in table 2. The most common postdementia discharge diagnosis was chronic renal failure, accounting for 27% of the admissions among those hospitalised, followed by disorders of the urinary system (21%). When readmissions for the same cause were excluded, the most common causes of hospitalisation were disorders of the urinary system, pneumonia and fracture of the femur, accounting for 15%, 10% and 6% of admissions, respectively (online supplementary table 1). Considering primary discharge diagnoses at ICD-10 chapter level, the highest proportions of admissions

**Table 1** Descriptive characteristics of analysis population

| | Dementia severity | | | |
| --- | --- | --- | --- | --- |
| | Mild | Moderate | Severe | All |
| | n=2057 | n=2616 | n=545 | n=5218 |
| Mean age (SD) | 81.7 (6.7) | 82.6 (7.1) | 82.4 (7.5) | 82.2 (7.0) |
| Sex, N (%) | | | | |
| Male | 765 (37.2) | 926 (35.4) | 189 (34.7) | 1880 (36.0) |
| Female | 1292 (62.8) | 1690 (64.6) | 356 (65.3) | 3338 (64.0) |
| Ethnicity, N (%) | | | | |
| Afro-Caribbean | 219 (10.6) | 400 (15.3) | 87 (16) | 706 (13.5) |
| Asian | 87 (4.2) | 105 (4.0) | 30 (5.5) | 222 (4.3) |
| European | 1683 (81.8) | 2015 (77.0) | 394 (72.3) | 4092 (78.4) |
| Other | 68 (3.3) | 96 (3.7) | 34 (6.2) | 198 (3.8) |
| Hospitalisation (12 months after diagnosis) | | | | |
| Number of patients with ≥1 inpatient admission (%) | 986 (47.9) | 1328 (50.8) | 282 (51.7) | 2596 (49.8) |
| Number of patients per 100 py (95% CI)* | 54.6 (51.3 to 58.1) | 60.0 (56.9 to 63.3) | 63.3 (56.4 to 71.2) | 58.1 (55.9 to 60.4) |
| Including readmissions for the same cause | | | | |
| Number of hospitalisations† | 2413 | 3201 | 664 | 6278 |
| Mean number of admissions (SD) | 2.4 (5.9) | 2.4 (5.9) | 2.3 (4.6) | 2.4 (5.8) |
| Median (IQR) inpatient stay | 2 (1 to 10) | 3 (1 to 13) | 4 (1 to 14) | 3 (1 to 12) |
| After excluding readmissions for the same cause | | | | |
| Number of hospitalisations† | 1868 | 2461 | 508 | 4837 |
| Mean number of admissions (SD) | 1.9 (1.4) | 1.9 (1.3) | 1.8 (1.2) | 1.9 (1.3) |
| Median (IQR) inpatient stay | 3 (1 to 12) | 5 (1 to 15) | 6 (2 to 17) | 4 (1 to 14) |

*P value= 0.097 ($\chi^2$ test).
†P value= 0.0001 (Kruskal-Wallis tests).
py, person years.

were observed in specific diseases of the genitourinary, circulatory and respiratory systems, accounting for 54%, 25% and 22% of the hospitalisation episodes, respectively (table 3); however, the broader chapters for 'other symptoms/signs' and 'injury/external cause events' were also common, accounting for 35% and 22% of admissions, respectively. After exclusion of repeat admissions (online supplementary table 2), the highest proportions of admissions at a chapter level were in specific diseases of the genitourinary, circulatory and respiratory systems, each accounting for around 20% of the hospitalisation episodes. The broader chapters for 'other symptoms/signs' and 'injury/external cause events', accounted for 31% and 21% of admissions, respectively, after exclusion of repeat admissions at a three-digit ICD level.

SARs for each discharge diagnosis by the severity of dementia at diagnosis are also shown in tables 2 and 3, and are illustrated in figures 2 and 3. For the cohort as a whole, patients with dementia were hospitalised more than the catchment population (SAR significantly greater than 1) for most of the 20 most common discharge diagnoses, the exceptions being chronic renal failure (N18), other chronic obstructive pulmonary disease (J44), heart failure (I50, SARs not significantly >1) and other cataracts (H26, SAR <1). The trends across

dementia severity groups were positive for 17 of the 20 conditions (increased SARs with increasing severity, indicated by a positive regression coefficient); however, only the trend for the R41 code ('other symptoms and signs involving cognitive function and awareness') was statistically significant. Considering ICD chapters, in most groups, patients with dementia were hospitalised more than the catchment population (table 3). The exceptions were cancers, benign neoplasms and diseases of the eye (SAR <1). Of the 18 ICD-10 chapters, trends across dementia severity groups were positive in 10; however, only those for mental and behavioural disorders (F), diseases of the respiratory system (J) and diseases of the genitourinary system (N) were statistically significant. No chapters showed significantly decreasing SARs with increasing dementia severity, although the negative trend for diseases of the digestive system (K) was close to significance. After excluding repeat admissions for the same cause, similar trends were observed at the three-digit ICD and chapter level, although an increasing trend of SAR with dementia severity reached or approached statistical significance for disorders of the urinary system and pneumonia with an increasing trend with dementia severity, and a decreasing trend, which reached statistical significance, for pain in throat

**Table 2** SARs by primary discharge diagnosis (20 most common ICD-10 three-digit code) for hospitalisations in the 12-month period following a dementia diagnosis

| Three-digit ICD primary discharge diagnosis | Number of hospitalisations with this discharge diagnosis | Proportion (%) of episodes with this discharge diagnosis | SAR (95% CI) by dementia severity at diagnosis | | | | Regression coefficient* | P value* |
|---|---|---|---|---|---|---|---|---|
| | | | All cases | Mild n=986 | Moderate n=1328 | Severe n=282 | | |
| Chronic renal failure (N18) | 710 | 27.3 | 1.0 (0.9–1.0) | 0.8 (0.7–0.9) | 1.0 (0.9–1.1) | 2.2 (1.9–2.6) | 0.71 | 0.26 |
| Other disorders of urinary system (N39) | 532 | 20.5 | 2.7 (2.5–2.7) | 2.4 (2.1–2.8) | 2.9 (2.6–3.2) | 4.8 (3.9–5.8) | 1.18 | 0.21 |
| Pneumonia, organism unspecified (J18) | 263 | 10.1 | 1.9 (1.7–2.2) | 1.4 (1.1–1.8) | 2.1 (1.8–2.5) | 4.5 (3.5–5.7) | 1.56 | 0.2 |
| Fracture of femur (S72) | 156 | 6.0 | 2.4 (2.0–2.8) | 2.4 (1.8–3.1) | 2.3 (1.9–2.9) | 3.1 (1.9–4.6) | 0.32 | 0.39 |
| Senility (R54) | 155 | 6.0 | 2.9 (2.5–3.4) | 2.9 (2.2–3.7) | 2.8 (2.2–3.5) | 4.6 (3.0–6.7) | 0.86 | 0.36 |
| Syncope and collapse (R55) | 133 | 5.1 | 2.9 (2.4–3.4) | 3.0 (2.3–3.9) | 2.8 (2.2–3.6) | 4.6 (2.9–6.9) | 0.77 | 0.4 |
| Other cataract (H26) | 132 | 5.1 | 0.6 (0.5–0.7) | 0.5 (0.4–0.7) | 0.6 (0.5–0.8) | 0.6 (0.3–1.0) | 0.05 | 0.58 |
| Unspecified acute lower respiratory infection (J22) | 103 | 4.0 | 1.8 (1.5–2.2) | 1.4 (1.0–2.0) | 1.9 (1.5–2.5) | 3.9 (2.5–5.8) | 1.24 | 0.21 |
| Cerebral infarction (I63) | 100 | 3.9 | 1.7 (1.4–2.1) | 1.4 (1.0–2.0) | 1.8 (1.3–2.3) | 3.4 (2.1–5.2) | 1.00 | 0.23 |
| Other symptoms and signs involving cognitive function and awareness (R41) | 96 | 3.7 | 5.2 (4.2–6.3) | 4.2 (2.9–6.1) | 5.9 (4.5–7.7) | 8.1 (4.6–13.2) | 1.93 | 0.05 |
| Pain in throat and chest (R07) | 91 | 3.5 | 1.4 (1.1–1.7) | 1.7 (1.2–2.3) | 1.3 (1.0–1.8) | 1.2 (0.5–2.4) | −0.24 | 0.16 |
| Other chronic obstructive pulmonary disease (J44) | 87 | 3.4 | 1.0 (0.8–1.2) | 1.0 (0.7–1.4) | 1.0 (0.8–1.4) | 0.8 (0.3–1.6) | −0.13 | 0.32 |
| Open wound of head (S01) | 78 | 3.0 | 3.0 (2.4–3.8) | 3.2 (2.1–4.5) | 2.8 (2.0–3.9) | 4.5 (2.4–7.7) | 0.65 | 0.47 |
| Unspecified dementia (F03) | 77 | 3.0 | 8.3 (6.6–10.4) | 5.0 (2.9–7.9) | 10.6 (7.9–14.0) | 12.7 (6.8–21.7) | 3.87 | 0.17 |
| Superficial injury of head (S00) | 73 | 2.8 | 3.6 (2.8–4.5) | 3.6 (2.4–5.3) | 3.3 (2.3–4.5) | 6.9 (3.9–11.2) | 1.64 | 0.39 |
| Heart failure (I50) | 61 | 2.3 | 0.8 (0.6–1.1) | 0.6 (0.4–1.0) | 1.0 (0.7–1.4) | 0.8 (0.3–1.7) | 0.08 | 0.76 |
| Acute renal failure (N17) | 61 | 2.3 | 2.1 (1.6–2.8) | 1.6 (1.0–2.6) | 2.2 (1.5–3.1) | 5.7 (3.3–9.2) | 2.05 | 0.25 |

**Table 2** Continued

| Three-digit ICD primary discharge diagnosis | Number of hospitalisations with this discharge diagnosis | Proportion (%) of episodes with this discharge diagnosis | SAR (95% CI) by dementia severity at diagnosis | | | | | |
|---|---|---|---|---|---|---|---|---|
| | | | All cases | Mild n=986 | Moderate n=1328 | Severe n=282 | Regression coefficient* | P value* |
| Alzheimer's disease (G30) | 60 | 2.3 | 10.4 (8.0–13.4) | 7.0 (3.9–11.5) | 10.8 (7.4–15.2) | 31.9 (19.5–49.2) | 12.46 | 0.24 |
| Other functional intestinal disorders (K59) | 59 | 2.3 | 1.8 (1.4–2.3) | 2.6 (1.8–3.6) | 1.5 (1.0–2.2) | 0.9 (0.2–2.5) | −0.85 | 0.09 |
| Pneumonitis due to solids and liquids (J69) | 52 | 2.0 | 3.0 (2.2–3.9) | 1.7 (0.8–3.0) | 3.5 (2.4–5.0) | 7.9 (4.4–13.1) | 3.14 | 0.15 |

*The regression coefficient indicates the increase or decrease in SAR associated with a one-step increase in severity of dementia. See Statistical analyses section for the linear model relating to these variables. The p value is for a two-sided test of the null hypothesis regression coefficient=0.

ICD-10, International Classification of Diseases, 10th edition; SARs, standardised admission ratios.

and chest (online supplementary tables 3–4, figures 2 and 3).

## DISCUSSION

In this retrospective cohort study of people with a dementia diagnosis, over half had at least one hospitalisation in the 12 months after diagnosis. The risk of hospitalisation and the LOS increased with higher dementia severity, although risk of hospitalisation did not reach statistical significance. Most of the common reasons for hospitalisation, based on the primary discharge diagnosis, were also more common than expected in cases relative to the catchment population. Differences in SARs by baseline dementia severity, where present, were more evident for discharge diagnoses grouped at ICD-10 chapter level than at the level of specific three-digit ICD-10 codes.

The hospitalisation rate for at least one inpatient admission in the 12-month period after dementia diagnosis was 58 per 100 py, slightly higher than reported elsewhere.[21 22] This may reflect more recent data (given that hospitalisation rates are increasing), the near-complete data on hospitalisation in our study and/or the fact that hospitalisation is free at the point of delivery in the UK context.[7] Increasing likelihood and LOS with increasing dementia severity is consistent with other reports, although differences between admissions from community and institutional facilities have been described, with reduced healthcare use following relocation to institutional care, potentially reflecting higher support.[23–25] This, as well as survival effects, may account for our finding that the most marked difference in hospitalisation was between mild and moderate dementia rather than between moderate and severe dementia.

The most common causes of hospital admission, after excluding repeat admissions for the same cause, were disorders of the urinary system and pneumonia, followed by fracture of femur, senility and syncope and collapse; this is similar to the findings reported elsewhere.[4 9 26–28] Most of the common causes of admission identified in people with dementia were more frequent than expected relative to the catchment population (SAR >1).

Higher risk of syncope and collapse, and of other diagnoses related to fractures and falls, has been previously reported in patients with dementia,[4 9 26] and may reflect age-related autonomic dysfunction, comorbid disorders and polypharmacy,[29] impaired visuospatial functions and gait instability.[30] A review of published clinical trials of atypical antipsychotic medication use did not find evidence of increased injury, falls or syncope associated with their use,[31] nor did a recent study of hospitalised falls/fractures in a CRIS-derived cohort of people with dementia.[32] Significant predictors included social/demographic factors, physical health problems and previous episode; no associations were found with neuropsychiatric symptoms, cognitive (MMSE) scores or functional problems.[32] Inpatient admissions due to infection were also more common than expected, both at the level of specific disorders and at the ICD-10 chapter level. This may reflect more rapid age-related decline in immune function,[33] problems with mobility associated with urinary incontinence and subsequent infection[34] or inactivity and dysphagia causing respiratory tract infection.[35] Acute renal failure is reported to be associated with a high risk of hospital readmission.[36 37] In our study it remained a common cause of admission among individuals with dementia even after excluding repeat admissions. Risk factors for acute renal failure include hospitalisation,

**Table 3** SARs (95% CI) by discharge diagnosis (ICD-10 chapter) for hospitalisations in the 12-month period after a dementia diagnosis

| Discharge diagnosis by ICD-10 chapter | | Number of hospitalisations with this discharge diagnosis | Proportion (%) of episodes with this discharge diagnosis | SAR (95% CI) by dementia severity at diagnosis | | | | Regression coefficient* | P>|T|* |
|---|---|---|---|---|---|---|---|---|---|
| | | | | All cases | Mild n=986 | Moderate n=1328 | Severe n=282 | | |
| A | Infectious and parasitic diseases | 62 | 2.4 | 1.5 (1.2–2.0) | 1.0 (0.6–1.7) | 1.8 (1.3–2.5) | 2.1 (0.9–3.9) | 0.51 | 0.16 |
| B | Infectious and parasitic diseases | 11 | 0.4 | 1.6 (0.8–2.9) | 0.8 (0.1–2.8) | 2.1 (0.8–4.2) | 2.8 (0.3–10.0) | 0.99 | 0.11 |
| C | Cancers | 167 | 6.4 | 0.3 (0.3–0.4) | 0.3 (0.3–0.4) | 0.4 (0.3–0.5) | 0.1 (0.1–0.3) | −0.1 | 0.47 |
| D | Benign neoplasms or diseases of the blood | 158 | 6.1 | 0.8 (0.7–1.0) | 0.9 (0.7–1.1) | 0.9 (0.7–1.1) | 0.3 (0.1–0.7) | −0.3 | 0.31 |
| E | Endocrine, nutritional and metabolic diseases | 138 | 5.3 | 1.9 (1.6–2.3) | 1.9 (1.5–2.5) | 1.9 (1.4–2.4) | 2.1 (1.2–3.5) | 0.09 | 0.54 |
| F | Mental and behavioural disorders | 194 | 7.5 | 5.7 (4.9–6.6) | 4.7 (3.6–6.1) | 6.1 (5–7.4) | 7.4 (4.9–10.8) | 1.35 | 0.01 |
| G | Diseases of the nervous system | 183 | 7.0 | 2.4 (2.1–2.8) | 2.1 (1.6–2.7) | 2.4 (2.0–3.0) | 3.4 (2.3–5.0) | 0.65 | 0.18 |
| H | Diseases of the eye | 209 | 8.1 | 0.6 (0.5–0.7) | 0.6 (0.5–0.8) | 0.7 (0.5–0.8) | 0.3 (0.2–0.6) | −0.13 | 0.43 |
| I | Diseases of the circulatory system | 638 | 24.6 | 1.3 (1.2–1.3) | 1.4 (1.2–1.5) | 1.2 (1.1–1.4) | 1.0 (0.8–1.4) | −0.16 | 0.06 |
| J | Diseases of the respiratory system | 567 | 21.8 | 1.6 (1.5–1.6) | 1.3 (1.1–1.5) | 1.7 (1.5–1.9) | 2.2 (1.8–2.7) | 0.46 | 0.03 |
| K | Diseases of the digestive system | 448 | 17.3 | 1.0 (0.9–1.0) | 1.2 (1.1–1.4) | 1.0 (0.8–1.1) | 0.6 (0.4–0.9) | −0.3 | 0.06 |
| L | Diseases of the skin | 105 | 4.0 | 1.2 (1.0–1.5) | 1.2 (0.9–1.6) | 1.1 (0.8–1.4) | 1.9 (1.1–3.0) | 0.33 | 0.41 |
| M | Diseases of the musculoskeletal system | 239 | 9.2 | 0.9 (0.8–1.1) | 1.1 (0.9–1.3) | 0.8 (0.7–1.0) | 0.9 (0.6–1.4) | −0.07 | 0.64 |
| N | Diseases of the genitourinary system | 1389 | 53.5 | 1.3 (1.2–1.3) | 1.1 (1.0–1.1) | 1.4 (1.3–1.4) | 1.7 (1.4–1.9) | 0.28 | 0.01 |
| R | Symptoms and sign not elsewhere classified | 913 | 35.2 | 1.9 (1.8–1.9) | 2 (1.8–2.2) | 1.9 (1.7–1.9) | 1.7 (1.3–2.1) | −0.15 | 0.18 |
| S | Injury, poisoning and certain other external causes | 574 | 22.1 | 2.3 (2.1–2.3) | 2.3 (2.0–2.7) | 2.2 (2.0–2.5) | 2.4 (1.9–3.1) | 0.05 | 0.69 |
| T | Injury, poisoning and certain other external causes | 120 | 4.6 | 1.6 (1.3–1.9) | 1.6 (1.2–2.1) | 1.8 (1.4–2.3) | 0.7 (0.2–1.5) | −0.46 | 0.47 |
| Z | External causes | 161 | 6.2 | 1.1 (1.0–1.3) | 1.6 (1.3–1.9) | 0.9 (0.7–1.1) | 0.8 (0.4–1.4) | −0.4 | 0.25 |
| Total | | 2596 | – | 1.3 (1.2–1.3) | 1.2 (1.2–1.2) | 1.3 (1.3–1.3) | 1.3 (1.2–1.3) | 0.03 | 0.3 |

*The regression coefficient indicates the increase or decrease in SAR associated with a one-step increase in severity of dementia. See Statistical analyses section for the linear model relating to these variables. The p value is for a two-sided test of the null hypothesis regression coefficient=0.

ICD-10, International Classification of Diseases, 10th edition; SARs, standardised admission ratios.

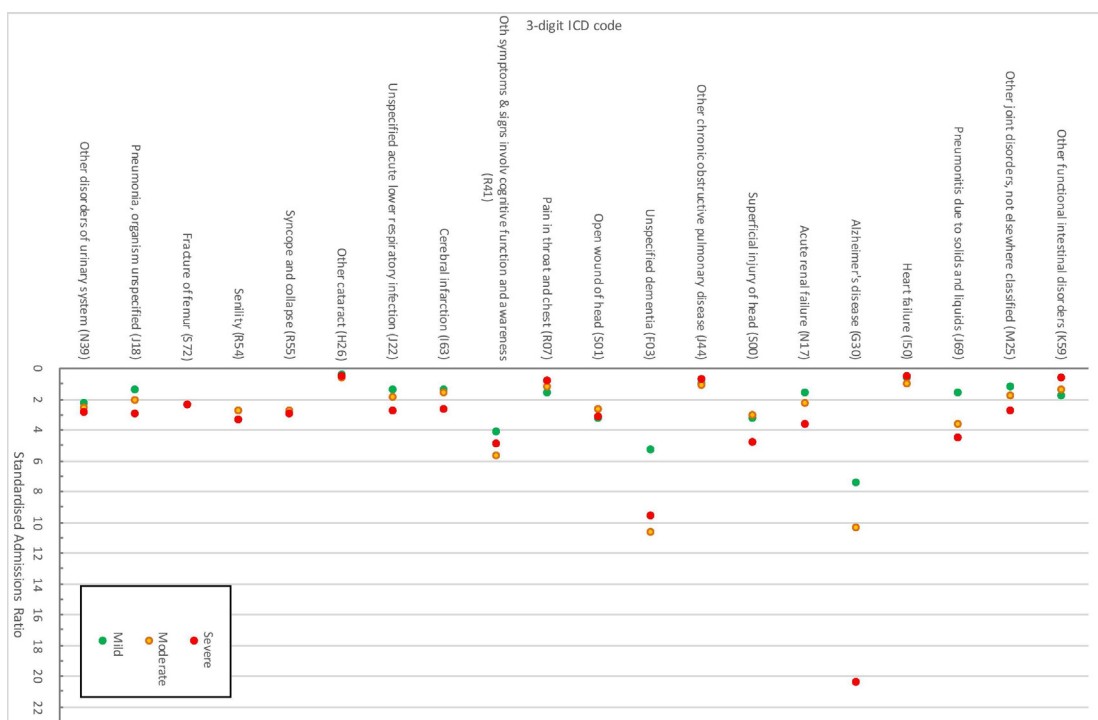

**Figure 2** SARs by three-digit ICD code for mild, moderate and severe dementia populations (excludes repeat admissions for the same cause at a three-digit ICD level).ICD, International Classification of Diseases; SARS, standardised admission ratios.

infection, diabetes, pre-existing renal impairment and advanced age,[38] which themselves may be associated with dementia severity.

A lower than expected frequency of hospitalisation (SAR <1) was observed for heart failure, cancers, benign neoplasms and diseases of the eye, consistent with other reports[21 28] and possibly reflecting diagnostic delay. For example, an association of dementia with later-stage cancer at diagnosis has been reported,[39] as well as unexpected cases being identified at autopsy[40]; furthermore, patients with dementia may be less likely to undergo invasive diagnostic tests and receive fewer treatment

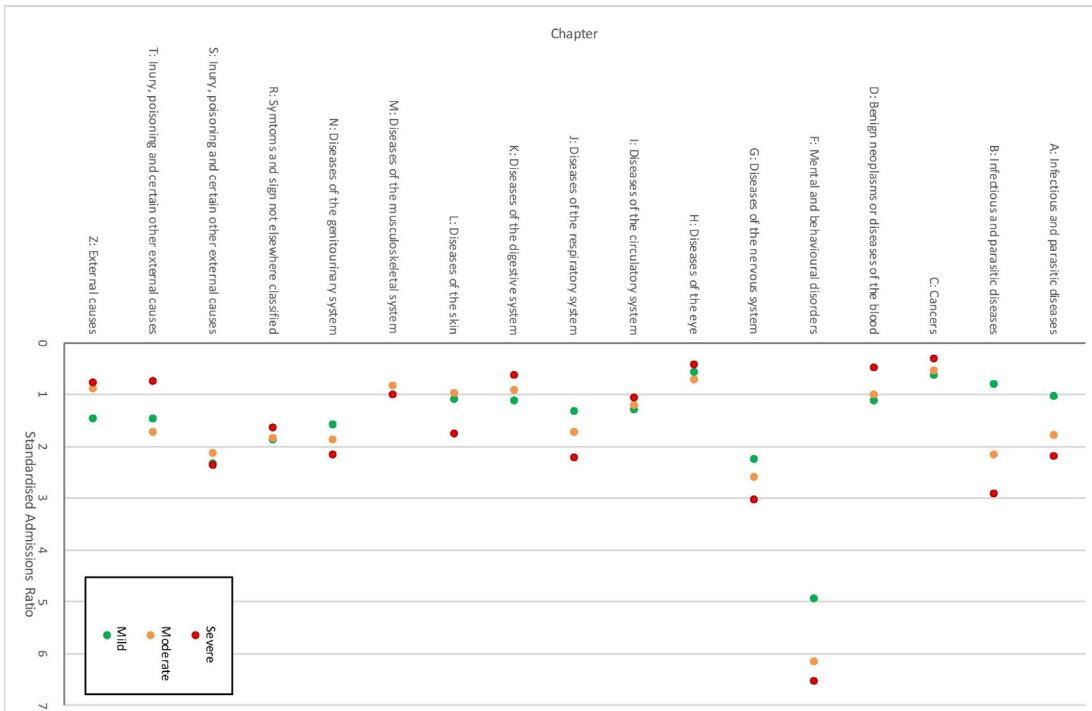

**Figure 3** SARs by ICD chapter for mild, moderate and severe dementia populations (excludes repeat admissions for the same cause at a three-digit ICD level). ICD, International Classification of Diseases; SARs, standardised admission ratios.

interventions, potentially explaining a shorter duration of survival.[41]

While rates of healthcare utilisation and costs, or limited specified causes of admission,[23 24] have been reported in relation to severity of dementia, we are not aware of any evaluation of common causes of hospital admission by dementia stage. Most of the SARs for specific causes of hospitalisation were positively associated with dementia severity, although most not to a significant extent. Associations, where significant, were more often with discharge diagnoses categorised by ICD-10 chapter rather than three-character code. This may have arisen because of higher statistical power for the chapter-level comparisons but might also reflect differences in the accuracy and/or variability of code assignments in routine administrative data; for example, there may be differences between hospitals in which specific three-character code to assign to a particular hospitalisation profile but stronger agreement on the broader chapter heading.

As well as hospitalisations occuring as a direct result of dementia, with worsening cognitive impairment there may be an increase in admissions to acute care for exacerbation, because of an individual's decreasing ability to manage existing comorbidity. Similarly, if identification of new comorbidity is delayed at earlier stages of dementia, this may increase the risk of hospitalisations later in its course due to underrecognised and/or undertreated disorders, although there is also evidence for an increased risk of some causes of hospitalisation even prior to a dementia diagnosis.[42] An increased LOS is consistent with the higher rate of complications seen in hospitalised patients with dementia, as well as with increasing severity of dementia.[43] Increasing respiratory and genitourinary disease admissions most likely represent infections, and diseases of the nervous system may reflect cases where the dementia itself has been provided as a primary diagnosis, or else comorbidities such as depression, psychosis or other behavioural/psychological manifestations. Risk for diseases of the digestive system diminished across the severity groups, but this may reflect a diminished ability to communicate non-specific symptoms rather than a reduced risk of defined disorders, as indicated by the negative coefficient for 'other functional intestinal disorders' (K59) in table 2, which was the most common digestive system diagnosis.

Strengths of the study included the large and representative sample of diagnosed cases of dementia and the linkage to a national hospitalisation database providing near-complete outcome ascertainment. Patients who were receiving care from acute hospital liaison services at the time of their dementia diagnosis were excluded from the study since these may reflect individuals whose diagnosis might have been precipitated by the hospitalisation, and yet who would also have an increased risk of rehospitalisation by virtue of their status, thus biassing the association of interest. Considering limitations, the analysed sample was from a single service and also only included cases with dementia diagnosed in specialist services; however,

estimated proportions of people with dementia in the SLaM catchment who receive a specialist diagnosis is relatively high at 75.2%, compared with 67.6% nationally.[44] This study was descriptive in nature and used all available data; as such it was not generated with a specific power calculation for the associations of interest, and the relatively small number of cases with severe dementia at diagnosis will have limited the detection of differences across the three groups. Furthermore, unrecognised dementia cases omitted from observed admission rates will have biassed findings towards the null, as would healthy survivor effects (which may also be particularly an issue for the severe dementia group). We used the discharge diagnosis recorded during the last episode of the hospital spell, and it is possible that the primary discharge diagnosis code may reflect complications that arose during the hospital stay rather than the reason for initial hospital admission. Although we compared SARs across dementia severities for a given discharge diagnosis, due to potential differences in the underlying age structure between severity categories SARs are not strictly comparable with each other. However, as the mean age was broadly similar between dementia severities, this mitigates this concern in part. Finally, this analysis did not attempt to account for comorbidity, medication use, institutional residence or socioeconomic status, did not subclassify dementia and was not able to investigate or take into account clustering by admission units.

Understanding the factors influencing health service utilisation by patients with dementia is necessary to inform care needs and to guide future healthcare resource planning and allocation. Our findings are of importance given that hospitalisation is a significant element in the cost of dementia care and that the prevalence of dementia is increasing. Our study highlights the need to develop specific strategies for those causes of hospitalisation that are most amenable to prevention in dementia. Further research on factors influencing patterns of healthcare use over time and severity would be useful in this context.

**Acknowledgements** The authors would like to acknowledge Hitesh Shetty for his contributions to the study.

**Contributors** UG was responsible for the design, analysis, interpretation and drafting of the manuscript. GP contributed to the design and analysis of the study. NWG provided statistical input and contributed to the drafting of the manuscript. RS contributed to the design, interpretation and drafting of the manuscript.

**Funding** This work was supported by NIHR Biomedical Research Centre at the South London and Maudsley NHS Foundation Trust and King's College London. RS is additionally part funded by a Medical Research Council Mental Health Data Pathfinder Award to King's College London, and an NIHR Senior Investigator Award.

**Disclaimer** The funder of the study had no role in the study design, data collection, data analysis, data interpretation or writing of the report. UG, GP and RS had access to the anonymised data. The corresponding author had the final responsibility for the decision to submit for publication.

**Competing interests** UG and NWG are employees of GSK, hold stock and receive a salary from GSK. RS has received research funding in the last 5 years from Roche, Janssen, Takeda and GSK.

**Patient and public involvement** Patients and/or the public were not involved in the design, or conduct, or reporting or dissemination plans of this research.

**Patient consent for publication** Not required.

**Provenance and peer review** Not commissioned; externally peer reviewed.

**Data availability statement** Data may be obtained from a third party and are not publicly available. Because of their nature and to comply with their ethical approval, CRIS data are required to remain within the firewall of the South London and Maudsley NHS Foundation Trust (SLaM). Access to the data used for this study can be facilitated by the CRIS Oversight Committee on application and with appropriate SLaM affiliation, details of which can be obtained from cris.administrator@slam. nhs.uk.

**ORCID iDs**
Usha Gungabissoon http://orcid.org/0000-0002-2040-1763
Gayan Perera http://orcid.org/0000-0002-3414-303X

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
