## [Reviewer comments · BMJ Open]

ARTICLE DETAILS

TITLE (PROVISIONAL)	The association between dementia severity and hospitalisation profile in a newly assessed clinical cohort: The South London and Maudsley Case Register
AUTHORS	Gungabissoon, Usha; Perera, Gayan; Galwey, Nicholas; Stewart, Robert

VERSION 1 – REVIEW

REVIEWER	Joanne Tropea Melbourne Health and University of Melbourne, Australia
REVIEW RETURNED	09-Dec-2019

GENERAL COMMENTS	Thank you for the opportunity to review this paper. In order for this paper to be accepted I am recommending major revision. The biggest issue for me is I don't feel the authors present a compelling enough argument for why severity of dementia is an important factor in frequency, LOS or cause of hospitalisations. What would this mean/how useful would this information be in the clinical setting? Introduction Very brief introduction - needs more. I am also not sure if the study is powered enough to look at 3 dementia categories, and the association with the outcomes of interest as this information is not provided. Methods section: It looks like elective hospital admissions as well as emergency admissions have been included. Can you please confirm? It isn't clear to me if you compared differences in likelihood of, LOS or reason for hospitalisation between mild and severe dementia. Can you please clarify? Results: Table 1: suggest adding SD to mean values and % to number values. I would have liked to see total number of hospitalisations on this table too. Can you please provide the n(%) for these categories. Table 2: please add the word "primary" before diagnosis in the title I'm not sure about including the table on ICD-10 chapters. Has this been done before in comparative studies? I would have thought to target treatment that would prevent these admissions a more detailed diagnosis would be more relevant.
---

	Discussion: I suggest adding the reference B Draper et al 2011 Inter Psychogeriatrics https://doi-org.ezp.lib.unimelb.edu.au/10.1017/S1041610211001694 In this Australian study they looked at hospitalisations of over 20,000 people living with dementia using linked dataset.
--	---

REVIEWER	Bernhard Michalowsky Germany Center for Neurodegenerative Diseases
REVIEW RETURNED	10-Dec-2019

GENERAL COMMENTS	This retrospective observational cohort study addresses the important and interesting topic of the frequency and causes of hospitalizations in the year following a dementia diagnosis using a huge sample/ database. Even though this paper is well written, I have some major comments that should be addressed to more clearly understand the results of this paper. Abstract  - Please add the numbers to the statement about the likelihood of admission and its association with dementia severity. Same for the most common discharge diagnoses. - There is nothing stated in the method section about the comparison between dementia patients and controls, nothing about used models to assess these differences (but I am a bit uncertain if really models were used if results were solely descriptive. - The conclusion does not come from the demonstrated results. Please revise. Introduction:  - The authors state in line 24, that “a few studies have evaluated the extent to which the risk and cause” without any reference. Please add all references, studies as well as study results. I think the introduction could benefit from a more in-depth look at all studies related to this topic. - Please more clearly state the main objective of this study. Methods  - The study design isn't really a study design. I would suggest combining both sections' design and setting. - Please clarify which dementia diagnoses were used to identified patients living with dementia (line 12 page 7) - I really don't understand why the authors use age- and sex-standardized admission ratios for the logistic regression model. I would suggest assessing the association between admission and having dementia stage adjusted for all other variables (age, sex, and so on). Furthermore, please refer to the association and not to the relationship. - I was wondering about whether or not the hospital wards should be handled as cluster und thus included as fixed or random effects in the multivariate models. Please clarify. - Page 10 Line 8. The “flow chart” of the identification of participants should be replaced in the method section. Results:  - The authors stated in the method section that “Age- and sex-standardized admission ratios - (SARs) with 95% confidence intervals (CI) were calculated for each of the 20 most frequent - causes of hospital admission for the 12-month period from the dementia diagnosis based on
--

	- the three-digit ICD-10 codes.”. However, in the results section, the authors present a regression confident of the association between each diagnosis a cognitive impairment. Please clarify in the method section how this analysis/ model was run. - Probably I missed it, but in the abstract, the authors stated that PwD was more often admitted as the “catchment population”, but I cannot find this information in one of the Tables. I think, whether or not patients living with dementia were more often admitted to hospitals compared to healthy controls considering different discharge diagnoses would be very interesting. Limitation: - There is some evidence, that the utilization of healthcare services, like hospitals, is higher in the year before receiving the dementia diagnoses and subsequently decrease in the year after. This topic could be included in the discussion and limitation section. - https://doi.org/10.1016/j.jalz.2015.07.149
--	---

REVIEWER	Umberto Maggiore Dipartimento di Medicina e Chirurgia Università di Parma Parma, Italy
REVIEW RETURNED	21-Dec-2019

GENERAL COMMENTS	The study is interesting, and well written. An intriguing finding is that the most common post-dementia discharge diagnosis was CKD. That may be caused by competing risk: , elderly patients not dying because of cardiac disease, or cancer survive long enough to develop dementia and CKD. Moreover there is a non-statistically significant numerical association between discharge diagnosis of acute renal failure AKI (which s related to CKD) and dementia severity. It has been shown that hospital-acquired AKI experience higher risk of early hospital readmission. (eg Koulouridis et al AJKD 2015, Silver Am J Med 2017). This phenomenon might be amplified by the association between dementia severity and urinary tract infections. Therefore, dementia severity, CKD, AKI, and urinary tract infections may interact with each other in increasing the RE-hospitalization rates Few minor statistical remarks Ordinary least sqaures regression treats each outcome value as having the same statistical precision (i.e. assumes that the varance of each SAR is the same) whereas the variance of SAR for severe dementia is much larger compared to moderate dementia. Poisson regression - with expected counts as offset var, or weightet least squares may be more appropriate (although I am not sure whether in this setting that would make a difference). Moreover, personally I would have regarded the analysis perfomed after excluding repeated admission as my primary analysis: in my view, a primary analysis using repeated admissions shoud take into account of the fact the data are repeated within subject. Moreover, Fig S1, and S2 (which exclude repeat admissions) are excellent as they allow the visual appraisal of quite a large number of estimates. I would consider to move them to the main body of the manuscript.
---

REVIEWER	Chen, SC Institute of Medicine and School of Medicine, Chung Shan Medical University, Taichung, TAIWAN
REVIEW RETURNED	04-Jan-2020

GENERAL COMMENTS	Statistical Review: The main aim of this study was to examine the association between the causes of hospitalisations following a dementia diagnosis and the severity of dementia by using standardised admission ratios and regression models. These statistical methods in this manuscript were appropriate. Because the standardised admission ratio was based upon an indirect fashion of standardization, the authors should address the limitations of this method in the discussion section.
---

REVIEWER	Paul Desan MD PhD Yale School of Medicine USA
REVIEW RETURNED	07-Jan-2020

GENERAL COMMENTS	Gungabissoon and colleagues study hospital admissions in the 12 months after a new diagnosis of dementia, using a national hospitalization database. The most common primary discharge diagnoses and ICD chapter level diagnoses are presented, in absolute terms and as standardized admission ratios, and the effect of dementia severity on these outcomes studied. The analysis and presentation are competent and thorough. The strength of the study is the completeness of the database and the rigorous analysis, and the weakness is the use of one service provider only. This is an area where many studies have been published, and the addition of relationship to dementia severity does not generate any novel new insights. While dementia is an important clinical issue, this new study does not help us design specific approaches to avoiding hospital admissions. The editors will need to weigh interest to the average BMJ Open reader in comparison to other submissions to the journal. Perhaps the study would be better directed to a journal with a more geriatric or epidemiological focus.
--

VERSION 1 – AUTHOR RESPONSE

Reviewer(s)' Comments to Author:

Reviewer: 1

In order for this paper to be accepted I am recommending major revision. The biggest issue for me is I don't feel the authors present a compelling enough argument for why severity of dementia is an important factor in frequency, LOS or cause of hospitalisations. What would this mean/how useful would this information be in the clinical setting?

RESPONSE: We thank the reviewer for the time taken to review and comment on our work. We have provided more context in the introduction, outlining the evidence gaps and potential clinical application of our work (justification of the study). Although we have discussed potential reasons why severity of dementia is an important factor with regards to hospitalisation patterns, we agree with the reviewer's suggestion, and we have provided additional wording in the discussion. We feel that the association

of interest is primarily of importance for further research – for example, interventions to prevent unnecessary hospitalisations need to understand any differences in admission profiles by dementia stage in order to be tailored appropriately.

Introduction

Very brief introduction - needs more.

RESPONSE: We thank the reviewer for their comment. As mentioned, we have provided more background that we feel is relevant for contextualising our study.

I am also not sure if the study is powered enough to look at 3 dementia categories, and the association with the outcomes of interest as this information is not provided.

RESPONSE: We have added a sentence in the discussion section stating the following: "This study was descriptive in nature and used all available data; as such it was not generated with a specific power calculation."

Methods section:

It looks like elective hospital admissions as well as emergency admissions have been included. Can you please confirm?

RESPONSE: We apologise that this was not mentioned; we have now clarified this in the methods section; both elective and unplanned admissions were included, which we felt was most policy-relevant.

It isn't clear to me if you compared differences in likelihood of, LOS or reason for hospitalisation between mild and severe dementia. Can you please clarify?

RESPONSE: We apologise for any lack of clarity. Hospitalisation likelihood and LOS data overall were displayed descriptively and were not initially formally compared between severity groups; instead the broad observation was made that SARs were very similar between severity groups, and that median inpatient stay durations were also similar. As a result of the reviewers feedback we have made some form comparisons between dementia severity with hospitalisation and length of hospital stay which have been added to the methods and results sections.

Results:

Table 1: suggest adding SD to mean values and % to number values. I would have liked to see total number of hospitalisations on this table too. Can you please provide the n(%) for these categories.

RESPONSE: We have updated Table 1 as requested.

Table 2: please add the word "primary" before diagnosis in the title

RESPONSE: This has been added

I'm not sure about including the table on ICD-10 chapters. Has this been done before in comparative studies? I would have thought to target treatment that would prevent these admissions a more detailed diagnosis would be more relevant.

RESPONSE: We accept the limitations to this approach, but would prefer these data to be in the public domain, as they provide a systems-level comparison that would be difficult to provide in any alternative form. Analyses of more detailed diagnostic codes are also provided; however, these in turn have the potential limitation of identifying single common conditions and not the cumulative smaller effects within systems that might be more apparent at chapter-level. We also have some concerns that administrative data might contain higher levels of error and heterogeneity at a more detailed diagnostic level (e.g. due to varying local practice on common specific codes) than they would at chapter level. We therefore believe that evaluating admissions at a ICD-10 chapter level, although broad, can provide some useful insights that would otherwise be lost when evaluating diagnoses at a 3-digit level. We have added text to the Discussion on this issue.

Discussion:

I suggest adding the reference B Draper et al 2011 Inter Psychogeriatrics <https://doi-org.ezp.lib.unimelb.edu.au/10.1017/S1041610211001694>
In this Australian study they looked at hospitalisations of over 20,000 people living with dementia using linked dataset.

RESPONSE: We have added this reference to the manuscript.

Reviewer: 2

This retrospective observational cohort study addresses the important and interesting topic of the frequency and causes of hospitalizations in the year following a dementia diagnosis using a huge sample/ database. Even though this paper is well written, I have some major comments that should be addressed to more clearly understand the results of this paper.

Abstract

- Please add the numbers to the statement about the likelihood of admission and its association with dementia severity. Same for the most common discharge diagnoses.

RESPONSE: We are grateful for these comments. We have incorporated the reviewer's suggestions to the abstract as requested

- There is nothing stated in the method section about the comparison between dementia patients and controls, nothing about used models to assess these differences (but I am a bit uncertain if really models were used if results were solely descriptive.

RESPONSE: We used indirect standardisation to estimate standardised admissions ratios to enable us to compare the observed to expected numbers of admissions amongst individuals with dementia compared to the general catchment population, after standardising by age and gender. We did not use any regression models to compare rates between cases and the general population. No updates to the abstract have been made, however further detail has been provided to the main methods section to clarify this.

- The conclusion does not come from the demonstrated results. Please revise.

RESPONSE: We thank the reviewer for flagging this to us; we have revised the conclusion accordingly.

Introduction:

- The authors state in line 24, that “a few studies have evaluated the extent to which the risk and cause” without any reference. Please add all references, studies as well as study results. I think the introduction could benefit from a more in-depth look at all studies related to this topic.

RESPONSE: We have added this to the introduction.

- Please more clearly state the main objective of this study.

RESPONSE: We have revised the wording in the manuscript to address this comment.

Methods

- The study design isn't really a study design. I would suggest combining both sections' design and setting.

RESPONSE: We have incorporated this suggestion within the manuscript.

- Please clarify which dementia diagnoses were used to identified patients living with dementia (line 12 page 7)

RESPONSE: We thank the reviewer for their comment; this has been added to the methods section.

- I really don't understand why the authors use age- and sex-standardized admission ratios for the logistic regression model. I would suggest assessing the association between admission and having dementia stage adjusted for all other variables (age, sex, and so on).

RESPONSE: As we hope is now clearer in the Introduction, we were specifically interested in investigating hospitalisation rates relative to the catchment population, as we believe that these have higher potential public health relevance and communicability than an analysis carried out within a dementia cohort. We therefore focused on generating standardised admission ratios, in addition to the analyses carried out of SAR variation by dementia stage. Given the use of aggregate data for the catchment it would not have been possible to use a logistic regression approach to compare people with and without dementia. We have revised Introduction and Methods text and hope that our approach is better explained in this respect.

Furthermore, please refer to the association and not to the relationship.

RESPONSE: We have incorporated these changes.

- I was wondering about whether or not the hospital wards should be handled as cluster und thus included as fixed or random effects in the multivariate models. Please clarify.

RESPONSE: Unfortunately we do not have data on hospital wards to enable us to account for this. We have however added this as a limitation in the discussion.

- Page 10 Line 8. The “flow chart” of the identification of participants should be replaced in the method section.

RESPONSE: We have incorporated the reviewer's suggestion.

Results:

- The authors stated in the method section that “Age- and sex-standardized admission ratios (SARs) with 95% confidence intervals (CI) were calculated for each of the 20 most frequent causes of hospital admission for the 12-month period from the dementia diagnosis based on the three-digit ICD-10 codes.”. However, in the results section, the authors present a regression confident of the association between each diagnosis and cognitive impairment. Please clarify in the method section how this analysis/ model was run.

RESPONSE: We have added text in the Methods section to clarify this.

- Probably I missed it, but in the abstract, the authors stated that PwD was more often admitted as the “catchment population”, but I cannot find this information in one of the Tables. I think, whether or not patients living with dementia were more often admitted to hospitals compared to healthy controls considering different discharge diagnoses would be very interesting.

RESPONSE: We thank the reviewer for their comment. We have added this into the abstract and into table 3. As emphasised above, the relative likelihood of admission for given causes was calculated using SARs because of the composite nature of catchment data.

Limitation:

- There is some evidence, that the utilization of healthcare services, like hospitals, is higher in the year before receiving the dementia diagnoses and subsequently decrease in the year after. This topic could be included in the discussion and limitation section. <https://doi.org/10.1016/j.jalz.2015.07.149>

RESPONSE: We thank the reviewer for this suggestion and have included a sentence describing increased utilisation prior to diagnosis of dementia.

Reviewer: 3

The study is interesting, and well written.

An intriguing finding is that the most common post-dementia discharge diagnosis was CKD. That may be caused by competing risk: ,elderly patients not dying because of cardiac disease, or cancer survive long enough to develop dementia and CKD. Moreover there is a non-statistically significant numerical association between discharge diagnosis of acute renal failure AKI (which s related to CKD) and dementia severity. It has been shown that hospital-acquired AKI experience higher risk of early hospital readmission. (eg Koulouridis et al AJKD 2015, Silver Am J Med 2017). This phenomenon might be amplified by the association between dementia severity and urinary tract infections. Therefore, dementia severity, CKD, AKI, and urinary tract infections may interact with each other in increasing the RE-hospitalization rates

RESPONSE: We thank the reviewer for these comments and are glad that they agree that this is an interesting finding. We have provided additional wording and references describing the high risk of re-admissions seen in individuals with acute renal disease in the discussion.

Few minor statistical remarks

Ordinary least squares regression treats each outcome value as having the same statistical precision (i.e. assumes that the variance of each SAR is the same) whereas the variance of SAR for severe dementia is much larger compared to moderate dementia. Poisson regression - with expected counts

as offset var, or weighted least squares may be more appropriate (although I am not sure whether in this setting that would make a difference).

RESPONSE: We thank the reviewer for their comment. We do not believe any action is required regarding this comment, but please let us know if you disagree.

Moreover, personally I would have regarded the analysis performed after excluding repeated admission as my primary analysis: in my view, a primary analysis using repeated admissions should take into account of the fact the data are repeated within subject. Moreover, Fig S1 and S2 (which exclude repeat admissions) are excellent as they allow the visual appraisal of quite a large number of estimates. I would consider to move them to the main body of the manuscript.

RESPONSE: As suggested by the reviewer, we have moved figures S1-S2 to the main body of the manuscript. In order to maintain flow of the manuscript and the a priori approach to analyses, we would prefer to keep our original primary analysis as such; however appreciate the reviewer's comments.

Reviewer: 4

Statistical Review:

The main aim of this study was to examine the association between the causes of hospitalisations following a dementia diagnosis and the severity of dementia by using standardised admission ratios and regression models. These statistical methods in this manuscript were appropriate. Because the standardised admission ratio was based upon an indirect fashion of standardization, the authors should address the limitations of this method in the discussion section.

RESPONSE: We thank the reviewer for their input and comment on our study. We have incorporated their suggestions into the discussion.

Reviewer: 5

Gungabissoon and colleagues study hospital admissions in the 12 months after a new diagnosis of dementia, using a national hospitalization database. The most common primary discharge diagnoses and ICD chapter level diagnoses are presented, in absolute terms and as standardized admission ratios, and the effect of dementia severity on these outcomes studied. The analysis and presentation are competent and thorough. The strength of the study is the completeness of the database and the rigorous analysis, and the weakness is the use of one service provider only. This is an area where many studies have been published, and the addition of relationship to dementia severity does not generate any novel new insights. While dementia is an important clinical issue, this new study does not help us design specific approaches to avoiding hospital admissions. The editors will need to weigh interest to the average BMJ Open reader in comparison to other submissions to the journal. Perhaps the study would be better directed to a journal with a more geriatric or epidemiological focus.

RESPONSE: We appreciate the comments and, as described in response to Reviewer 1, have added text to the Introduction on the rationale for the study. We do appreciate that there has been other research in this area and hence chose to submit to BMJ Open on the basis of the journal's remit to consider primarily methodology and research ethics rather than impact. Having said this, we do believe that the question is worth investigating and that our findings, whilst null in many

circumstances, are relatively novel in this respect and provide at least a template for others to consider replication studies.

VERSION 2 – REVIEW

REVIEWER	Bernhard Michalowsky German Center for Neurodegenerative Diseases (DZNE), Germany
REVIEW RETURNED	10-Feb-2020

GENERAL COMMENTS	The authors incorporated all of my comments, I have nothing to add.
---

REVIEWER	Umberto Maggiore Dipartimento di Medicina e Chirurgia, Università di Parma
REVIEW RETURNED	16-Feb-2020

GENERAL COMMENTS	I have no further remarks
---------------------------